# The Effects of Pepper (*Zanthoxylum bungeanum*) from Different Production Areas on the Volatile Flavor Compounds of Fried Pepper Oils Based on HS-SPME–GC–MS and Multivariate Statistical Method

**DOI:** 10.3390/molecules27227760

**Published:** 2022-11-11

**Authors:** Wenjing Niu, Honglei Tian, Ping Zhan

**Affiliations:** 1College of Life Science and Technology, Tarim University, Alaer 843300, China; 2College of Food Engineering and Nutritional Science, Shaanxi Normal University, Xi’an 710100, China; 3Shaanxi Provincial Research Center of Functional Food Engineering Technology, Xi’an 710100, China; 4The Engineering Research Center for High-Valued Utilization of Fruit Resources in Western China, Ministry of Education, Shaanxi Normal University, Xi’an 710100, China

**Keywords:** fried pepper oils, HS-SPME–GC–MS, volatile flavor compounds, hierarchical cluster analysis, principal component analysis

## Abstract

Fried pepper oil retains the overall flavor outline of pepper, and its unique rich and spicy flavor is deeply loved by consumers. In order to study the effect of different production areas of pepper on the flavor compounds of fried pepper oil, taking dried pepper from seven different production areas as raw materials, and taking rapeseed oil as a carrier oil as well as a constant frying temperature to prepare pepper oil, the present study analyzed the volatile flavor components of pepper oil qualitatively and quantitatively by employing headspace solid phase microextraction (HS-SPME) and gas chromatography–mass spectrometry (GC–MS). The principal component analysis (PCA) method was used to construct the correlation analysis model of volatile flavor substances among different samples of pepper oil. Applying the hierarchical cluster analysis (HCA), the main volatile substances causing the flavor differences of pepper oil from different production areas were identified. The results showed that a total of 81 chemical components were identified, including 15 alcohols, 10 aldehydes, 5 ketones, 34 hydrocarbons, 11 esters, 6 acids, and others. Terpinen-4-ol, linalool, 2,4-decadienal, trans-2-heptenal, sabinene, linalyl acetate, bornyl acetate, myrcene, 1-caryophyllene, trans-α-ocimene, and limonene were selected as the main substances leading to the flavor differences among the pepper oil samples. These 11 chemical components played a decisive role in the construction of the overall aroma of the pepper oil. Using a descriptive sensory analysis, it was concluded that pepper oil from different production areas holds different aroma intensities. Compared with the other six samples, S4 Hanyuan Pepper Oil (HYPO) shows a relatively strong trend toward a spicy fragrance, fresh grassy fragrance, floral and fruity fragrance, fresh sweet fragrance, and fatty aroma.

## 1. Introduction

Pepper (*Zanthoxylum bungeanum*) belonging to the rutaceae *Zanthoxylum* genus, which can be divided into two categories of green and red pepper, is an aromatic spice unique to our country [1,2]. Throughout the ages, pepper has been used in all aspects of life, from medicine to cooking. As a kind of Chinese herbal medicine, pepper is pungent and warm and can be used to relieve abdominal pain and cold extremities caused by lack of yang energy and cold invasion in pain, with the effect of warming the middle and dispersing cold [3]. The active antioxidant ingredients in pepper, such as polyphenols, flavonoids and alkaloids, can play an antioxidant role by scavenging excess free radicals and inhibiting free radical production [4]. In addition, the active ingredients extracted from the volatile oil as well as the bioactive bases of pepper have strong anti-inflammatory and analgesic effects, which can effectively relieve pain caused by lumbar sprains, arthritis, and inflammation of dental pulp, etc., [3,5]. Pepper can also be used for external treatment for eczema and itching, which is related to the use of its volatile oil [6]. Modern pharmacology has found its inhibitory and destructive effects on dermatophytes and intestinal roundworms, etc., [6]. As a spice, it ranks first in the “thirteen incense”, and its unique strong spicy flavor has the effect of calming mutton [7] and removing a fishy smell [8], it can also inhibit the formation of carcinogenic heterocyclic amines during cooking [9]. The planting areas of pepper are mainly concentrated in East Asia [10,11], and the Sichuan Province, Yunnan Province, Chongqing, Shaanxi Province, Shanxi Province, and other places, are the main production areas of pepper in China [12]. The aroma substances in pepper are easily affected by the environmental factors in the production area, such as precipitation, soil quality and temperature, as well as the different varieties; therefore, pepper from different production areas has different compositions and contents of volatile oils and different peppers holds unique fragrances, thus there are differences in aroma, and also quality [13,14]. As one of the deep-processing products of pepper, pepper oil facilitates the consumption of pepper and preserves the complete flavor of the pepper [15]. The commercially available pepper oil has a strong pungent hemp-like taste, but the fried pepper possesses a strong salty flavor. In addition, fried pepper oil has a stronger aroma-emission performance and a higher sensory experience, which can better meet the public needs [16].

At present, domestic and foreign studies mainly focus on the volatile oil of peppernd the extraction of pepper oil, and there are few studies on the aroma substances of fried pepper oil. Liu et al. [17] separated volatile substances from pepper oil by HS-SPME and solvent-assisted flavor evaporation, and analyzed 32 kinds of odor active substances by gas chromatography–mass spectrometry olfactometry (GC–MS-O). SUN J. et al. [18] studied the aroma characteristics of Shaanxi Hancheng Pepper Oil (HCPO) and Sichuan Hanyuan Pepper Oil (HYPO) through an aroma-extract dilution analysis (AEDA) and a gas chromatography–mass spectrometry olfactometer (GC–MS-O). The odor activity values (OAVs) showed that the OAVS of 28 odor substances were ≥1 in HCPO or HYPO. By successfully stimulating the aroma characteristics of HCPO and HYPO through aroma recombination model, the study found that β-phellandrene, p-cymene, octyl acetate, octanoic acid, citronellol, and sabinene play key roles in the aroma difference between HCPO and HYPO. Ni Ruijie et al. [19] used GC–IMS combined with multivariate statistical method to analyze the volatile flavor compounds of four kinds of pepper oil prepared from different frying times, and selected 20 kinds of iconic differences, including linalool, β-pinene, α-pinene, limonene, α-phellandrene, hexanal, heptanal, and (E)-2-hexanal, etc. Chen Haitao et al. [16] used simultaneous distillation extraction and solvent-assisted evaporation to extract the volatile flavor components in fried pepper oil, and employed GC–MS for qualitative and quantitative analysis of the extract. The key aroma compounds in the extract were identified by the gradient dilution method combined with gas chromatography and olfactometry (GC–O). A total of 75 volatile flavor components were detected by the two methods. Linalool, germacrene-D, limonene, linalyl acetate, and 4-terpineol acetate were selected as the key aroma-active compounds in the fried pepper oil. 

According to the geographical origin and distribution of Chinese pepper, the author selected the most distinctive seven species of Chinese pepper, as shown in Figure 1. The current study takes pepper oil as the research object, and employs a headspace solid-phase microextraction (HS-SPME) and gas chromatography–mass spectrometry (GC–MS) to analyze the differences of volatile flavor compounds in pepper oil from the same process, from different production areas. Moreover, through descriptive sensory analysis, principal component analysis (PCA), and hierarchical cluster analysis (HCA), this study analyzes the differences in volatile components among the samples of pepper oil and selects out the iconic compounds, in order to initially explore the flavor composition of pepper oil and provide a theoretical basis for the creation of the flavor quality-control technology of pepper.

## 2. Results and Discussion

### 2.1. The GC–MS Test Analysis of Pepper Oil Samples

Using HS-SPME and GC–MS, the volatile flavor compounds of seven kinds of pepper oils were separated and identified. The results are shown in Table 1 and Figure 2.

As shown in Table 1, differences among the seven pepper oils were evident for the amount and the composition of volatiles. A total of 81 volatile flavor compounds were detected from seven samples of pepper oil by HS-SPME–GC–MS method, including 15 kinds of alcohols, 10 kinds of aldehydes, 5 kinds of ketones, 34 hydrocarbons, 11 types of esters, 6 types of acids, and others. Among them, the content of the alcohols, hydrocarbons, and esters was higher, followed by the ketones and aldehydes, and the lowest content was the acids. Samples S1~S7 contained 54, 53, 47, 49, 37, 19, and 16 volatiles, respectively. The common aroma components of the pepper oil from the seven different varieties were 5-isopropyl-2-methylbicyclo [3.1.0] hexane-2-alcohol, terpinen-4-ol, linalool, 2,4-decadienal, myrcene, sabinene,1-caryophyllene, α-caryophyllene, and trans-α-ocimene. Among these, alkenes and alcohols played an important role in the aroma profile of pepper oils. However, some hydrocarbons, such as β-pinene and β-phellandrene, which have been identified as key aroma compounds in pepper oil [18], were not detected in our study. The lack of these compounds may be attributed to the different growth environments of pepper or the different detection conditions [9]. Some aldehydes were generated from unsaturated fatty acids through thermal oxidative degradation of fats, usually associated with a fried and fatty flavor, such as 2,4-decadienal and (E)-2-heptenal [20]. The aroma of the fried pepper oil is not represented by one or several chemical components: in addition to the main aroma compounds, it also includes compounds that play a coordinating role, a modifying role, and a fixing role. Some of these compounds have very little content. It may not be possible to evaluate the impact of a certain component on the aroma of the pepper oil by the relative content or aroma value alone. It is evident that there are significant differences in the composition of pepper from different varieties and geographical regions, which to some extent lead to the different aroma characteristics of pepper.

Figure 2 shows that the relative content of the alcohol chemical components in the seven kinds of pepper oils were relatively high, and there were seven kinds of compounds with relatively high alcohol contents, including: α-terpineol, cis-4-butynol, 5-isopropyl-2-methylbicyclo [3.1.0] hexane-2-ol, eucalyptol, terpinen-4-ol, linalool, and γ-Terpineol. The content of linalool is relatively high, up to 303.59 ug/g. Linalool is widely used in perfumery, and it has the fresh and sweet smell of flowers and woods, as well as an anti-oxidation and anti-aging effect, and a certain effect on the treatment of insomnia [21]. γ-Terpineol was detected in the S4 and S5 samples; however, the magnitude of the aroma contribution of Trans-4-Platycladusorientalis alcohol in pepper oil cannot be confirmed because it was only detected in the S4 and S5 oils, and needs to be further determined by OAV. This may be one of the reasons why S4 and S5 behaved differently from the other pepper oils in the HCA and PCA analyses. Trans-4-Platycladus orientalis alcohol was detected in the S4 and S5 samples, and this chemical component is mentioned in domestic research [22,23,24], so it should be an important chemical component that imparts the characteristic flavor of fried pepper oil components. The contribution of a substance to the overall aroma of the pepper oil cannot be determined by its relative content alone; the contribution of different aroma substances to the aroma of the pepper oil will vary when the threshold value and the concentration in the sample matrix are different, and the magnitude of the contribution of a certain aroma component to the overall aroma of the sample cannot be accurately determined by its content alone; only the components with higher aroma values constitute the characteristic aroma of the pepper oil. A total of 10 aldehydes were identified, and the relative content was low. There are three kinds of compounds with a relatively high content of aldehydes, including: trans-2,4-decadienal, 2,4-decadienal, and trans-2-heptenal. The threshold value of aldehydes is relatively low, and their contribution to the aroma of fried pepper oil cannot be ignored. Aldehydes provide a fresh fragrance and woody aromas, which are particularly similar to botanical aromas. The saturated fatty aldehydes (n-hexanal and nonanal) extracted by HS-SPME can impart fruity and floral aromas to the samples. A total of five kinds of ketones have been identified, and there were three kinds of compounds with a relatively high content of ketones, including 4-(1-methylethyl)-2-cyclohexen-1-one, 2,3-dihydrogen-3,5-ihydrodxy-6-methyl-4H-pyran-4-one, and piperonone. Piperonone has the cool smell of mint and camphor, and it is a natural spice. A total of 34 kinds of hydrocarbons were identified, which have a large relative content. There are 13 kinds of compounds with a relatively high content of hydrocarbons, including myrcene, α-phellandrene, limonene, sabinene, (*E*)-3,7-dimethyl-1, 3,6-octatriene, γ-terpinene, 1-caryophyllene, α-caryophyllene, β-elemene, trans-α-ocimene, α-pinene, camphene, and trans-β-ocimene. Hydrocarbon compounds are the main aroma compounds in pepper, which mainly give the fresh grassy fragrance and the pungent flavor of fried pepper oil. A total of 11 kinds of esters have been identified, and there were 5 kinds of compounds with a relatively high content of esters, including linalyl acetate, neryl acetate, terpinyl acetate, bornyl acetate, and 1,5-dimethyl- 1-Vinyl-4-hexyl-2-aminobenzoate. Esters mainly reflect aromas such as fruity, woody, and grassy aromas [25]. Linalyl acetate and neryl acetate have the aromas of tropical fruits. Linalyl acetate can provide floral aroma, which is the main characteristic flavor component in fried pepper [26]. Bornyl acetate has the smell of wood, camphor, and mint. A total of six kinds of acids and other compounds were identified, and there were two kinds of compounds with a relatively high content in this category, including: acetic acid and *Zanthoxylin*. *Zanthoxylin* belongs to the amide compound, showing numbness, and it is one of the characteristic compounds of pepper [27]. 

According to the Guadagni theory of odor activity values, the components of food with high aroma concentrations and low threshold values are likely to be the characteristic aromas of the food. A high concentration of substances does not necessarily contribute significantly to the aroma of pepper oil, while low levels of substances may also have a large impact on the overall aroma of pepper oil. Therefore, to judge whether the volatile matter contributes to the aroma components of *Zanthoxylum bungeanum* oil, we should not just look at the content of volatile matter. The contents of various compounds in different varieties of pepper oil are different. Due to the different production areas, geographical and climatic conditions, maturity, air-drying methods, fertilization factors, and applicable processing methods of different varieties of pepper, the components of aroma substances of pepper oil may be affected.

### 2.2. Sensory Analysis of Pepper Oil Samples from Different Production Areas

From the flavor radar chart in Figure 3, it can be seen that when compared with the other six samples, S4 (HYPO) shows a relatively strong trend toward a spicy fragrance, fresh grassy fragrance, floral and fruity, fresh sweet fragrance, and fatty aroma. It may be that the higher relative content of limonene and linalool in the S4 sample is relevant. According to the literature, linalool has the fragrance of flower, green, wood and sweet, and limonene has the fragrance of citrus and light balsam [28]. The difference in quality of pepper is determined to a certain extent by the content of limonene and linalool [29]. The aroma reorganization of the volatile flavor substances of pepper oil constitutes the overall flavor profile, and the aroma contribution of different volatiles is different, depending not only on the amount of volatile content, but also on the odor threshold and the results of the interaction between aroma substances. These substances play a great role in differentiating the overall aroma and quality differences of pepper oil from the different origins [30,31], which is consistent with the research results of Gao Xiajie [14]. Figure 2 shows that the relative content of hydrocarbon compounds in HYPO is the largest among the seven kinds of pepper oils, and this chemical component mainly imparts the fresh grassy fragrance and spicy aroma of fried pepper oil, which has a certain contribution to the overall flavor of pepper. S3 (GGPO), S7 (JJPO), and S6 (PCPO) all showed a weakening trend in the four sensory attributes of spicy fragrance, fresh grassy fragrance, fresh sweet fragrance, and fatty fragrance. The fragrance is very weak, but can be distinguished; S1 (HCPO), S2 (YSPO), and S5 (MCPO) have a spicy fragrance, fresh grassy fragrance, fresh sweet fragrance, and a fatty fragrance. There are small fluctuations in the four sensory attributes: the pungent, green, sweet, and fatty fragrances, and the aromas are easy to feel and their intensity increases significantly. The sensory attributes of S6 and S7 were weaker in terms of a green and spicy flavor, as there is a link between the smaller variety of the volatile flavor compounds contained in these two types of pepper oils, especially the smaller variety of hydrocarbons, which mainly provide the green and spicy flavor of fried pepper oil and are the main flavor-presenting compounds in pepper as well as in pepper by-products. The description officer assessed that the two kinds of green pepper oil had a stronger smell than the red pepper oil. Since linalool was the main factor affecting the smell of the pepper, it can be seen from Table 1 that the content of linalool in S6 and S7 was higher. The trend of the flower and fruit aroma in S4 and S5 is obvious, which is related to the content of linalyl acetate in the two, thus providing a floral fragrance and the limonene intensity of a lemon fruit fragrance in harmony. In terms of a floral and fruity fragrance, except for the S4 sample, the other six samples showed large fluctuations, showing that the aroma intensity was S4 (HYPO), S5 (MCPO), S7 (JJPO), S2 (YSPO), S6 (PCPO), S3 (GGPO), and S1 (HCPO). The aroma intensity of the different varieties of pepper oil is different when evaluating its the sensory attributes as it can be easily affected by the variety of the pepper tree, the sunshine hours at the planting area, the soil quality, the rainfall, and the temperature due to the volatile flavor substances.

### 2.3. Cluster Analysis of Volatile Components in Pepper Oil

In order to clarify the key different substances among the seven kinds of pepper oils, a cluster analysis was carried out on the various volatile flavor substances of the seven kinds of pepper oil samples. The results are shown in Figure 4.

Figure 4 shows that the seven kinds of pepper oil samples can be distinguished well by cluster analysis. The size of the Euclidean distance can characterize the similarity of the samples, and the samples with a greater similarity, that is, with a smaller Euclidean distance, can be classified into one category [32]. Seven different varieties of pepper oil can be divided into three groups: the first group has three varieties of samples 1, 2, and 3; the second group has two varieties of samples 6 and 7; and the third group has two samples of the 4 and 5 varieties. Among them, the maximum difference of the Euclidean distance of the first groups of pepper oil samples is about 12, and S1 and S2 form clusters within the minimum distance, indicating that these two groups of samples are very similar. When the inter-class distance reaches 25, all the samples of pepper oil can be classified into one class. Therefore, the volatile flavor compounds of different varieties of pepper oil samples can be distinguished by HCA analysis. Hanyuan pepper is mainly produced in the Hanyuan County, Sichuan Province, which has a warm winter and cool summer, four distinct seasons, cold uplands, hot river valleys, low and uneven rainfall, and vertical climate changes; the unique climate and ecological environment has nurtured the unique flavor and excellent quality of Hanyuan pepper. Mao County pepper is produced in the Mao County, Sichuan Province, with unique natural conditions and distinct regional characteristics. Mao County is located in the Min River arid river valley, with a dry climate, low annual precipitation, large temperature differences between day and night, and sufficient sunshine; Mao County pepper production has good natural geographic and climatic conditions. S4 and S5 are produced from the Sichuan Province, with similar environments of origin and belong to the same variety of red pepper oil, so the environmental conditions and the group formed in Figure 4 have some correlation. S6–S7 belong to different production areas of green pepper oil; S1, S2, and S3 all belong to the same variety of red pepper oil produced from different areas. Therefore, the results of the analysis of volatile components are related to the variety of the pepper tree, the place of origin, the sunshine hours of the planting area, the soil quality, the rainfall and the temperature, the storage and processing after picking, the analysis methods, and other factors of the materials involved; hence, the aroma intensity of the flavored oil of the different varieties and the different production areas of pepper oil are different in the cluster analysis. 

According to Table 1, the contents of terpinen-4-ol (4.27, 4.84, 4.03 ug/g), linalool (24.04, 55.62, 58.36 ug/g), 2,4-decadienal (3.12, 2.86, 2.54 ug/g), trans-2-heptenal (1.27, 1.71, 1.37 ug/g), sabinene (13.31, 12.12, 12.96 ug/g), and linalyl acetate (27.09, 28.72, 26.63 ug/g) are similar in S1, S2, and S3; bornyl acetate (3.35, 3.6, 2.53 ug/g), limonene (38.59, 34.27, -ug/g), and 1-caryophyllene (5.05, 4.69, 2.35 ug/g) not only have similar contents in S1, S2, and S3, and these substances improve the similarity among the three samples, but also there are some differences in the contents of S1, S2, and S3. Terpinen-4-ol (1.61, 2.4 ug/g), linalool (198.24, 303.59 ug/g), 2,4-decadienal (1.73, 1.94 ug/g), myrcene (3.03, 4.63 ug/g),1-caryophyllene (2.64, 3.29 ug/g), and trans-α-ocimene (1.57, 2.71 ug/g) in S6 and S7 were very close, and these substances improve the similarity between the two samples and make the samples fall into one category over the shortest distance. Moreover, the contents of the four substances linalyl acetate, bornyl acetate, trans-2-heptenal, and limonene, in S6 and S7 are particularly low, and the existing instruments cannot even detect the levels of these substances. Therefore, they are very different from the other five samples, which further shows that these two samples are similar. The contents of terpinen-4-ol (5.59, 5.35 ug/g), linalool (115.64, 130.36 ug/g), 2,4-decadienal (6.94, 5.58 ug/g), trans-2-heptenal (3.24, 2.16 ug/g), limonene (57.89, 42.87 ug/g), 1-caryophyllene (5.68, 5.85 ug/g), linalyl acetate (298.2, 273.88 ug/g), and bornyl acetate (5.79, 3.82 ug/g) in S4 and S5 are very similar, compared with the other five samples; the latter four chemical components have higher contents in S4 and S5. These substances improve the similarity between S4 and S5 and make the two samples fall into one category over the shortest distance.

### 2.4. Principal Component Analysis of Pepper Oil

The principal component analysis of the content of each chemical component in different pepper oil samples is shown in Figure 5.

In the PCA analysis, PC1 = 47.88%, PC2 = 30.90%, and the cumulative variance contribution rate of the two principal components was 78.78%, which could better represent sample information. Therefore, the first two principal components (PC1–PC2) were selected for analysis. As seen in Figure 5, there were no duplicates between all the data points in samples S1–S7, indicating that the principal component analysis was able to identify and classify the samples, and the different production areas of the pepper seasoning oils were clustered in different regions of the PCA score plot. HYPO S4 and MCPO S5 were located in the upper right region of the coordinate axis, but the distance between the two samples was large, indicating that there are certain differences in the types and contents of their chemical components. HCPO S1, YSPO S2, and GGPO S3 were located in the lower right area of the score plot, indicating that the three samples had some similarity, and the distance interval between S1 and S2 samples was small, that is, they had an extremely high similarity. PCPO S6 and JJPO S7 were in the upper left area of the score map, and the distance between S6 and S7 was small, which indicated that S6 and S7 had great similarity. This result is consistent with the HCA results (Figure 4). It is evident that the principal component analysis method has a good differentiation effect on pepper oil from different origins, indicating that the relative contents of the volatile components in pepper oil from different origins have some degree of difference.

## 3. Materials and Methods

### 3.1. Materials and Instruments

There are seven varieties of pepper in this study: Dahongpao pepper from the Hancheng County, Shaanxi Province; Dahongpao pepper from the Yongshan County, Yunnan Province; Dahongpao pepper from the Gangu County, Gansu Province; Hanyuan Red pepper Oil from the Sichuan Province; Dahongpao pepper from the Mao County, Sichuan Province; Green pepper from the Pingchang County, Sichuan Province; and Green pepper from the Jiangjin District, Chongqing (all commercially mature, non-damaged dried pepper). These peppers were purchased from the Xinhong farmers’ market, Chenghua District, Chengdu City, Sichuan Province, China“Golden dragon fish Brand” rapeseed oil was from Changji City, Changji Hui Autonomous Prefecture, Xinjiang Uygur Autonomous Region, China. The N-alkane mixed standard (C7-C20),1,2-dichlorobenzene (chromatographically pure), was from Sigma-Aldrich, St. Louis, MO, USA.

The instruments were: 5977A-7890B Gas Chromatography–Mass Spectrometry (GC–MS) Agilent company, Santa Clara City, California, USA; Headspace Solid-Phase Microextraction (HS-SPME) Manual Sampling Device, 50/30 μm DVB/CAR/PDMS Extraction Head, Supelco company, Bellefonte, Pennsylvania, USA; HP -INNOWAX chromatographic column Agilent company, Santa Clara City, California, USA BS224S one ten thousandth balance Mettler Toledo company, Dover Delaware USA; and DNP-9272 oil bath pot Shanghai Jinghong Experimental Equipment Co., Ltd., Fengxian District, Shanghai, China.

### 3.2. Research Methods

#### 3.2.1. Preparation of Pepper Oil [33]

The following process outlines the pepper oil preparation: first, remove the impurities in the pepper, weigh 100 g pepper and 300 mL rapeseed oil with a 1/10,000 scale, then set the temperature parameter of the oil bath pot to 130 °C; pour the weighed rapeseed oil into a beaker and place it on the heating in the oil bath pot; after the temperature of the oil bath pot reaches the required level, fry the pepper at a constant temperature for 12 min; keep stirring during the frying process then stop the heating; cool down at room temperature; when the temperature drops to 40 °C, drain the peppercorns; separate the pepper oil; cool to room temperature and refrigerate for later use. Seven kinds of sequentially encoded pepper oils denoted as S1~S7, were obtained using the above-mentioned method from seven different pepper varieties, respectively, as follows: Hancheng pepper oil (HCPO), Yongshan pepper oil (YSPO), Gangu pepper oil (GGPO), Hanyuan pepper oil (HYPO), Mao county pepper oil (MCPO), Pingchang pepper oil (PCPO), and Jiangjin pepper oil (JJPO).

#### 3.2.2. Extraction of Volatile Flavor Substances

The extraction process is as follows: take 5.0 g of each pepper oil sample (S1~S7) and add it to a 15 mL headspace vial; add 0.1uL of standard 1, 2-dichlorobenzene as the internal standard compound; place the sealed headspace bottle in a water bath pot at 75 °C for 15 min; insert the aged extraction head into the headspace part of the sample bottle; adsorb it at the equilibrium temperature for 20 min; take out the adsorbed extraction fiber head and insert it into the GC–MS injection port immediately; desorb at 250 °C for 7 min; start GC–MS to collect data at the same time. Each experiment was conducted three times in parallel.

#### 3.2.3. Test Conditions of GC-MS

The GC conditions are as follows: HP-INNOWAX chromatographic columns (30 m × 0.25 mm, 0.25 μm); temperature of injection port: 250 °C; injection method: splitless injection; programmed temperature rise: keep at 50 °C for 5 min, raise the temperature to 80 °C at 5 °C/min, hold for 3 min, ramp up to 110 °C at 5 °C/min, hold for 3 min, ramp up to 140 °C at 5 °C/min, hold for 3 min, ramp up to 170 °C at 5 °C/min, hold for 3 min, then ramp at 5 °C/min to 200 °C, hold for 3 min, and finally to 230 °C at 5 °C/min, hold for 3 min; the carrier gas is helium; the flow rate is 1 mL/min. 

The MS conditions are: ion source temperature 230 °C; electron ionization source; electron energy 70 eV; transmission line temperature 280 °C; quadrupole temperature 150 °C; interface temperature 250 °C; mass scanning range 30–450 u, solvent delay 3 min.

#### 3.2.4. Quantitative and Qualitative Methods

The quantitative method involved: the internal standard method was used for quantification, and the relative content of each component was calculated according to the following Formula (1) through the ratio of the peak area of the internal standard to the peak area of each component in pepper oil:(1)$$C_{i}=f_{i}^{\prime }\times \frac{A_{i}}{A_{S}\times m}\times W_{S}$$

In the above formula: *C_i_* is the content of the component *i* to be measured, ug/g; *W_s_* is the quality of the added internal standard s, ug; *A_i_* and *A_s_* are the peak areas of the component *i* to be measured and the internal standard compound s to be tested, respectively; m is the quality of the sample to be tested, g; $f_{i}^{\prime }$ is the relative quality correction factor of the component *i* to be tested to the internal standard *s*; the relative correction factor of each component *i* to be tested in this experiment is 1.

The qualitative analysis involved: the isolated unknowns were matched with NIST/Wiley14 (only the identification results with a matching degree greater than 80% were selected); the relative retention index (*RI* value) of the related compounds in the reference was compared and analyzed. The compound retention index is calculated as the Formula (2):(2)$$RI=\left(\frac{\log t_{Ri}-\log t_{Rz}}{\log t_{R(z+1)}-\log t_{Rz}}+z\right)\times 100$$

In the above formula: *RI* is the retention index of the component to be tested; $t_{Rz}$ and $t_{R(z+1)}$ are the retention time of two n-alkanes immediately before and after aroma compound *i*; *Z* is the number of carbon atoms of n-alkanes; $t_{Ri}$ is the retention time of component *i* to be tested.

#### 3.2.5. Sensory Evaluation

The sensory evaluation is according to the following process. Take 10 mL of pepper oil and put it into seven 15 mL glass bottles. The liquid level of the pepper oil is at 2/3 of the glass bottle. Tighten the bottle stopper, take a water bath at 60 °C for 30 min, and quickly conduct the sensory evaluation when the temperature of pepper oil is in the range of 30~40 °C. Refer to the method of Meilgaard et al. [34] and Liu et al. [17] with appropriate modifications. In the present study, nine sensory evaluators were selected to form an evaluation team, and the evaluators were professionally trained to be familiar with the descriptors of evaluation and to use scales for evaluation. The reference standards for aroma are as follows: spicy fragrance (crushed spice), fresh grassy fragrance (the smell emitted after the grass is powdered), floral and fruity (1 mg/mL aqueous solution of phenylethyl alcohol), fresh sweet fragrance (the mixture of 100 mL aqueous solution containing 5 g sucrose and mint juice), fatty aroma (the aroma that escapes after the animal fat is heated and melted). These evaluate the pepper oil from the five sensory attributes of pepper oil (see the establishment of fuzzy mathematical model for details), and used a 5-point intensity scale, namely: 1 point: no aroma; 2 points: the fragrance is very weak, but can be distinguished; 3 points: it is easy to sense the fragrance; 4 points: the aroma intensity is obvious; 5 points: strong aroma. After evaluating one sample, the evaluators are required to gargle with clean water and evaluate the next sample after an interval of 10 min. After all evaluations are completed, the evaluation forms of the evaluation personnel are collected for subsequent statistical analysis.

The establishment of the fuzzy mathematical model [35]: taking the spicy fragrance, fresh grassy fragrance, floral and fruity, fresh sweet fragrance, and fatty aroma as the factor set, and expressing this as U, then U = {u1, u2, u3, u4, u5}. In addition, the method of Yang Liu et al. [35] and Zhu Youzhen et al. [36] is referred to for the distribution of the weight coefficients of each attribute in the sensory analysis, namely, the weight set X = {0.3, 0.2, 0.2, 0.2, 0.1}, where u1 is the spicy fragrance, u2 is the fresh grassy fragrance, u3 is floral and fruity, u4 is the fresh sweet fragrance, and u5 is fatty aroma.

#### 3.2.6. Data Analysis

In order to analyze the differences of the sensory characteristics of the intensity of the pepper oil, SPSS 25.0 software was used to analyze the variance of the evaluated data; PCA and HCA were processed by SPSS 25.0 software for data processing; Origin 2016 software was used for drawing. All data were subjected to three replicate experiments.

## 4. Conclusions

Descriptive sensory analysis and HS-SPME–GC–MS techniques were used to qualitatively and quantitatively analyze 81 aroma substances, including 15 alcohols, 10 aldehydes, 5 ketones, 34 hydrocarbons, 11 esters, and 6 acids and others, in pepper oil produced by the same process from seven different production areas as raw materials. The relative contents and types of volatile substances of the pepper oil from the different production areas were significantly different; however, the contribution of a certain aroma component to the overall aroma of the samples could not be accurately judged by its content alone, and only the components with higher aroma values had the characteristic aroma of pepper oil samples. Compared with the other six samples, HYPO S4 showed a relatively strong trend toward a spicy fragrance, fresh grassy fragrance, floral and fruity, fresh sweet fragrance, and fatty aroma. That is to say, the relative content of limonene and linalool in the S4 sample was relatively high, and to some extent, the quality of the pepper varieties was evaluated as good or bad. Cluster analysis classified the seven pepper-flavored oil samples into three major groups, with terpinen-4-ol, linalool, 2,4-decadienal, trans-2-heptenal, sabinene, linalyl acetate, bornyl acetate, myrcene, 1-caryophyllene, trans-α-ocimene, and limonene being the main substances responsible for the flavor differences among the pepper oil samples. These 11 chemical components played a decisive role in the construction of the overall aroma of the pepper oils. The aroma of the fried pepper oil is not formed by one or several volatile substances, but includes compounds that play a coordinating role, a modifying role, and a fixing role, in addition to the main aroma-contributing substances. Through the principal component analysis of pepper oil from seven different origins, the cumulative variance contribution of two of the principal components was 78.78%, and the differences among the different origins of pepper oil could be visualized from the principal component score plot. This method can distinguish pepper oil from different origins, and the established model was in good agreement with the relative content information of volatile substances in pepper oil. A variety of chemical components constitute the characteristic flavor of pepper, but only a few chemical substances contribute to the overall flavor. The volatile substances in pepper oil were analyzed by GC–MS, and these aroma substances are easily volatilized and even lost during production and processing, thus causing changes in the overall flavor profile of pepper oil. Furthermore, due to the continuous thermal reaction, the relative content of some chemical components decreases or increases, or new volatile substances are generated, which changes the natural flavor of pepper and forms the difference in aroma quality of different pepper oil samples. The data and analysis of this experiment can provide a reference for the future grade differentiation of pepper oil.

## Figures and Tables

**Figure 1 molecules-27-07760-f001:**
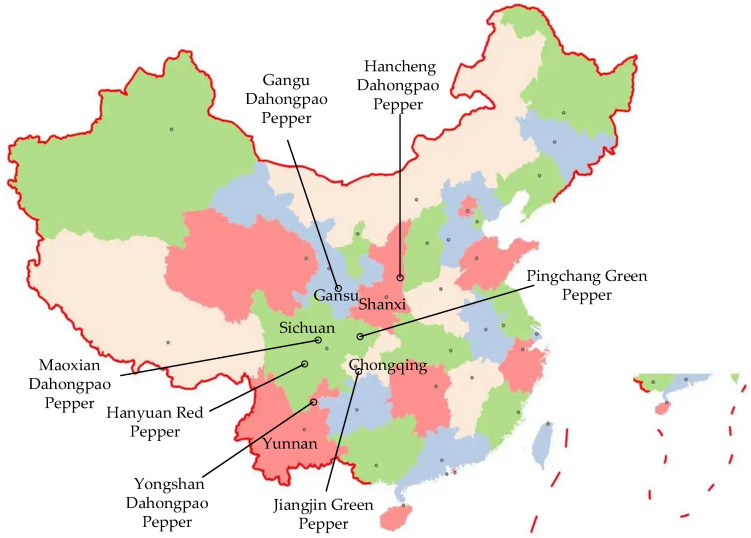
The geographical distribution of the seven most representative Chinese peppers.

**Figure 2 molecules-27-07760-f002:**
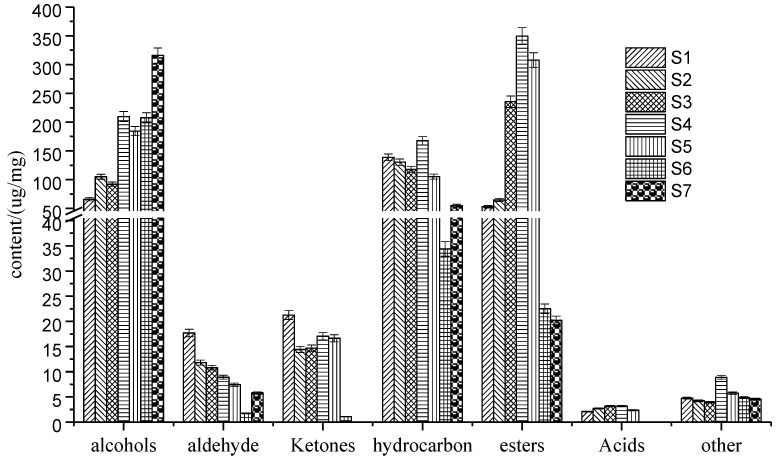
The comparison of the volatile components of seven kinds of pepper oil.

**Figure 3 molecules-27-07760-f003:**
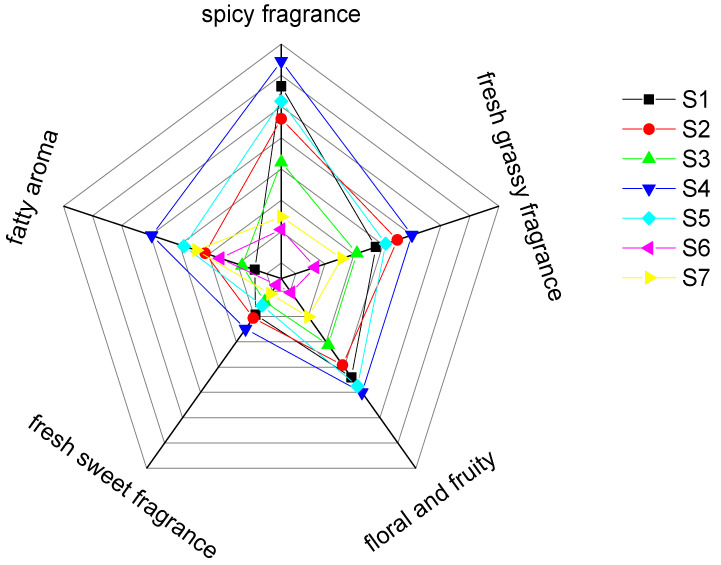
A comparison chart of flavor evaluation of the different varieties of pepper oil.

**Figure 4 molecules-27-07760-f004:**
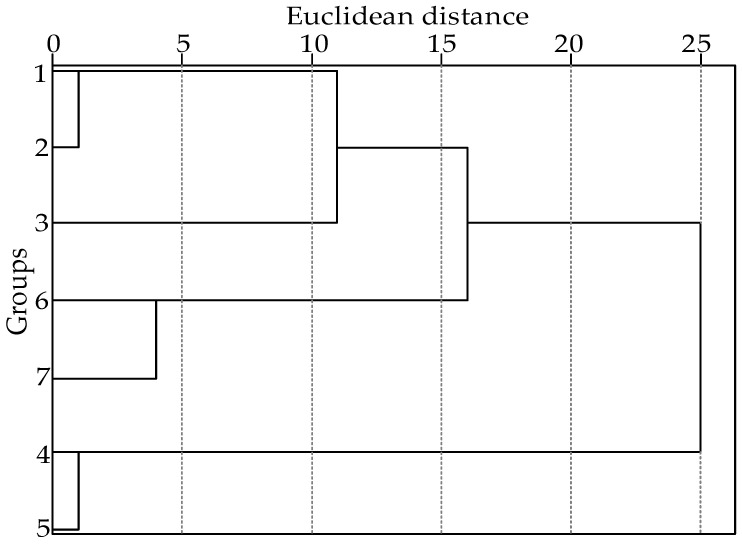
HCA of the volatile flavor substances in the seven kinds of pepper oil.

**Figure 5 molecules-27-07760-f005:**
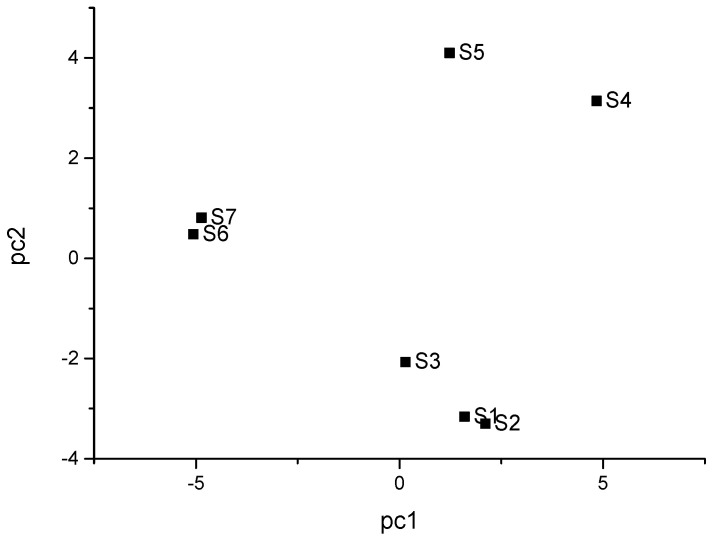
PCA score plots of the seven pepper oil samples.

**Table 1 molecules-27-07760-t001:** Aroma components and content of different varieties of pepper oils.

Retention Time (min)	Compound Name	*RI*/Literature Value	Identification Method	Compound Content (ug/g)						
				S_1_	S_2_	S_3_	S_4_	S_5_	S_6_	S_7_
Alcohols (15 kinds)										
17.069	α-Terpineol	1654/1662	*MS/RI*	12.43 ± 0.52	14.81 ± 0.62	7.52 ± 0.32	-	-	4.48 ± 0.19	5.12 ± 0.22
11.988	Cis-4-Butynol	1455/1450	*MS/RI*	-	-	1.61 ± 0.07	4.57 ± 0.19	4.1 ± 0.17	2.03 ± 0.09	-
13.39	5-Isopropyl-2-Methylbicyclo-3,1-Hexane-2-Ol	1472/1470	*MS/RI*	3.61 ± 0.15	5.55 ± 0.23	3.66 ± 0.15	2.84 ± 0.12	2.45 ± 0.1	1.36 ± 0.04	3.03 ± 0.13
10.433	Eucalyptol	1227/1220	*MS/RI*	18.1 ± 0.76	18.38 ± 0.77	15.46 ± 0.65	23.54 ± 0.99	-	-	-
16.548	Terpinen-4-Ol	1607/1612	*MS/RI*	4.27 ± 0.18	4.84 ± 0.2	4.03 ± 0.17	5.59 ± 0.24	5.35 ± 0.23	1.61 ± 0.07	2.4 ± 0.1
13.543	Linalool	1545/1556	*MS/RI*	24.04 ± 1.01	55.62 ± 2.34	58.36 ± 2.45	115.64 ± 4.8	130.36 ± 5.4	198.24 ± 8.3	303.59 ± 12.8
18.071	γ- Terpineol	1667/1669	*MS/RI*	-	-	-	40.74 ± 1.71	34.18 ± 1.44	-	-
15.883	2-Methyl-6-Heptene-3-Ol	1576/-	*MS*	0.62 ± 0.03	0.65 ± 0.03	-	-	-	-	-
27.321	Neroli Tert Alcohol	2038/2046	*MS/RI*	0.41 ± 0.02	-	-	-	-	-	-
16.188	Myrcenol	1593/-	*MS*	0.68 ± 0.03	0.69 ± 0.03	-	-	-	-	-
18.2	(1R,5S)-Rel- Carvacrol	1851/1858	*MS/RI*	0.41 ± 0.02	2.05 ± 0.09	-	-	-	-	-
17.061	Trans-4-Platycladusorientalis Alcohol	1638/1631	*MS/RI*	-	-	-	11.34 ± 0.48	8.1 ± 0.34	-	-
18.777	Trans Carvitol	1858/1860	*MS/RI*	1.56 ± 0.07	2.44 ± 0.10	1.57 ± 0.07	1.96 ± 0.08	-	-	-
29.654	Trans-Menthyl-2,8-Diene-1-Alcohol	1662/1670	*MS/RI*	-	-	-	3.45 ± 0.15	-	-	-
28.531	Geraniol	1847/1841	*MS/RI*	-	-	-	-	-	1.31 ± 0.06	1.65 ± 0.07
Aldehydes (10 kinds)										
21.109	Trans -2,4- Decadienal	1823/1826	*MS/RI*	-	4 ± 0.17	1.42 ± 0.06	3.86 ± 0.25	1.86 ± 0.08	-	-
22.119	2,4- Decadienal	1765/-	*MS*	3.12 ± 0.13	2.86 ± 0.13	2.54 ± 0.09	6.94 ± 0.29	5.58 ± 0.23	1.73 ± 0.07	1.94 ± 0.08
22.127	(E,E)-2,4-Dodecadienal	1803/-	*MS*	5.7 ± 0.24	-	4.52 ± 0.19	-	-	-	-
9.695	Trans-2,4-Heptadiene -Aldehyde	1389/1386	*MS/RI*	0.6 ± 0.03	0.7 ± 0.03	-	-	-	-	-
19.682	Trans-2-Decenylaldehyde	1634/1630	*MS/RI*	0.77 ± 0.03	0.52 ± 0.02	-	-	-	-	-
7.98	Trans-2-Heptenal	1312/1319	*MS/RI*	1.27 ± 0.05	1.71 ± 0.07	1.37 ± 0.06	3.24 ± 0.15	2.16 ± 0.07	-	-
15.659	Citronellal	1480/1478	*MS/RI*	0.66 ± 0.03	-	-	-	-	-	-
21.117	(E,E)-2,4- Decadienal	1756/1750	*MS/RI*	1.56 ± 0.07	1.53 ± 0.06	-	1.98 ± 0.08	-	-	-
8.854	n-Hexaldehyde	1335/1337	*MS/RI*	0.49 ± 0.02	0.65 ± 0.03	-	-	-	-	-
14.697	Nonanal	1452/1453	*MS/RI*	4.03 ± 0.17	2.7 ± 0.11	3.5 ± 0.15	-	-	-	-
Ketones (5 kinds)										
16.869	4-(1- Methyl Ethyl)-2-Cyclohexene-1-Ketone	1705/-	*MS*	5.54 ± 0.23	7.42 ± 0.31	3.46 ± 0.15	4.26 ± 0.18	3.33 ± 0.14	-	-
15.242	2,3-Dihydro-3,5-Dihydroxy-6-Methyl-4h-Pyran-4-Ketone	1692/-	*MS*	1.47 ± 0.06	4.74 ± 0.20	2.39 ± 0.1	3.21 ± 0.14	5.51 ± 0.23	-	-
19.354	Piperonone	1721/1743	*MS/RI*	10.53 ± 0.44	7.62 ± 0.33	6.5 ± 0.27	9.58 ± 0.4	7.81 ± 0.33	1.1 ± 0.05	-
18.953	Carvone	1708/1702	*MS/RI*	1.43 ± 0.06	-	0.99 ± 0.04	-	-	-	-
31.946	Pyranone	2249/2240	*MS/RI*	0.71 ± 0.03	1.09 ± 0.05	-	-	-	-	-
Hydrocarbons (34 kinds)										
9.062	Myrcene	1136/1143	*MS/RI*	20.54 ± 0.86	17.29 ± 0.73	12.95 ± 0.54	18.22 ± 0.77	10.94 ± 0.46	3.03 ± 0.13	4.63 ± 0.19
7.107	α-Phellandrene	1087/1086	*MS/RI*	4.76 ± 0.20	4.37 ± 0.18	5.96 ± 0.25	6.37 ± 0.23	-	-	-
11.138	Ocimene	1216/-	*MS*	6.44 ± 0.27	4.64 ± 0.2	3.68 ± 0.16	-	-	-	-
10.329	Limonene	1193/1196	*MS/RI*	38.59 ± 1.62	34.27 ± 1.44	-	57.89 ± 2.43	42.87 ± 1.8	-	-
8.501	Sabinene	1132/1130	*MS/RI*	13.31 ± 0.56	12.12 ± 0.51	12.96 ± 0.54	19.51 ± 0.82	10.36 ± 0.44	7.98 ± 0.34	13.45 ± 0.57
11.154	(E)-3,7- Dimethyl -1,3,6-Octriene	1241/1250	*MS/RI*	-	-	4.98 ± 0.21	4.92 ± 0.21	3.09 ± 0.13	-	-
9.88	α- Terpinene	1188/1194	*MS/RI*	-	-	-	5.56 ± 0.23	3.37 ± 0.14	-	-
11.579	γ-Terpinene	1251/1250	*MS/RI*	-	-	-	11.4 ± 0.48	7.06 ± 0.3	-	-
9.463	Phellandrene	1164/1160	*MS/RI*	-	-	1.04 ± 0.04	1.47 ± 0.06	-	-	-
29.269	γ- Cadinene	1758/1765	*MS/RI*	-	-	-	1.83 ± 0.08	1.62 ± 0.07	-	-
12.917	Terpinolene	1283/1271	*MS/RI*	2.9 ± 0.12	3.41 ± 0.14	-	-	-	-	-
25.069	Elemene	1583/1580	*MS/RI*	6.15 ± 0.26	4.66 ± 0.2	-	-	-	-	-
25.975	1- Caryophyllene	1596/1604	*MS/RI*	5.05 ± 0.21	4.69 ± 0.2	2.35 ± 0.1	5.68 ± 0.24	5.85 ± 0.25	2.64 ± 0.11	3.29 ± 0.14
27.089	α- Caryophyllene	1708/1706	*MS/RI*	1.42 ± 0.06	1.36 ± 0.06	1 ± 0.04	2.99 ± 0.13	3.44 ± 0.14	1.74 ± 0.07	2.15 ± 0.09
25.569	β- Elemene	1587/1580	*MS/RI*	-	-	1.94 ± 0.08	4.59 ± 0.19	3.79 ± 0.16	-	1.94 ± 0.08
10.721	Trans -α- Ocimene	1205/1218	*MS/RI*	13.78 ± 0.58	6.06 ± 0.25	7.81 ± 0.33	6.72 ± 0.28	3.52 ± 0.15	1.57 ± 0.07	2.71 ± 0.11
7.299	α- Pinene	1024/1028	*MS/RI*	-	2.94 ± 0.12	1.52 ± 0.06	2.7 ± 0.11	-	2 ± 0.08	-
23.506	Camphene	1495/1503	*MS/RI*	20.63 ± 0.87	17.62 ± 0.74	15.07 ± 0.63	-	-	-	-
10.168	1-Methyl-4-(1- Methyl ethyl)- Benzene	1190/-	*MS*	1.5 ± 0.06	1.58 ± 0.07	0.89 ± 0.04	1.7 ± 0.07	-	-	-
26.471	γ- Elemene	1602/1613	*MS/RI*	-	-	-	1.37 ± 0.06	1.42 ± 0.06	-	-
7.123	α-Limonene Thuene	1093/-	*MS*	-	-	-	9.23 ± 0.39	4.93 ± 0.21	-	-
14.705	Alloocimenols	1339/1334	*MS/RI*	-	-	-	2.1 ± 0.09	-	-	-
22.84	Germacrene-D	1482/1479	*MS/RI*	0.9 ± 0.04	0.93 ± 0.04	-	-	-	-	-
27.081	(+)-β- Selinene	1716/1729	*MS/RI*	-	-	1.16 ± 0.05	-	-	-	-
21.638	α- Piper Eggplant Olene	1447/1446	*MS/RI*	0.51 ± 0.02	0.5 ± 0.02	-	-	-	-	-
11.138	Trans -β- Basilene	1237/1230	*MS/RI*	-	9.02 ± 0.38	4.98 ± 0.21	4.92 ± 0.21	3.09 ± 0.13	-	-
14.697	(E,E)-2,6- Dimethyl -2,4,6-Octadiene	1323/-	*MS*	-	-	1.45 ± 0.06	-	-	-	-
26.463	3-Vinyl-2,5-Dimethyl-1,4-Hexadiene	1599/1601	*MS/RI*	0.68 ± 0.03	-	2.58 ± 0.11	-	-	-	-
25.974	Trans -β- Caryophyllene	1592/1595	*MS/RI*	-	-	-	4.96 ± 0.21	-	-	-
28.579	α-Claurolene	1728/1719	*MS/RI*	0.72 ± 0.03	0.6 ± 0.03	-	-	-	-	-
20.195	Vinyl Cyclohexene Dioxide	1428/-	*MS*	1.11 ± 0.05	1.1 ± 0.05	-	-	-	-	-
21.422	3-Carene	1443/1450	*MS/RI*	-	3.33 ± 0.14	-	-	-	-	-
10.459	M-Isopropyl Toluene	1201/1206	*MS/RI*	-	-	35.53 ± 1.49	-	-	17 ± 0.71	26.81 ± 1.13
28.026	Caryophyllene Oxide	1939/1935	*MS/RI*	-	-	-	4.53 ± 0.19	5.75 ± 0.24	-	-
Esters (11 kinds)										
24.828	Geranyl Acetate	1752/1746	*MS/RI*	-	-	2.83 ± 0.12	5.02 ± 0.21	3.44 ± 0.14	-	-
18.079	Linalyl Acetate	1546/1548	*MS/RI*	27.09 ± 1.14	28.72 ± 1.21	26.63 ± 1.12	298.2 ± 12.52	273.8 ± 11.5	-	-
24.115	Neryl Acetate	1728/1733	*MS/RI*	4.03 ± 0.17	2.35 ± 0.1	2.55 ± 0.11	3.96 ± 0.17	3.07 ± 0.13	-	-
23.69	Citronellyl Acetate	1657/1655	*MS/RI*	3.9 ± 0.16	2.75 ± 0.12	2.15 ± 0.09	3.43 ± 0.14	-	-	-
23.506	Terpinyl Acetate	1641/1652	*MS/RI*	-	-	-	28.57 ± 1.2	18.38 ± 0.77	-	-
23.89	Citronellyl Butyrate	1581/-	*MS*	-	-	2.38 ± 0.1	-	1.98 ± 0.08	-	-
31.321	4-Terpineol Acetate	1823/-	*MS*	-	-	-	-	-	6.75 ± 0.28	7.21 ± 0.3
21.422	3-Methyl-2-Cyclohexene-1-Ol-Acetate	1556/1557	*MS/RI*	-	-	-	4.65 ± 0.2	3.12 ± 0.13	-	-
23.193	Bornyl Acetate	1568/1566	*MS/RI*	3.35 ± 0.14	3.6 ± 0.15	2.53 ± 0.11	5.79 ± 0.24	3.82 ± 0.16	-	-
19.49	1,5-Dimethyl-1-Vinyl-4-Hexyl-2-Aminobenzoate	1552/-	*MS*	14.61 ± 0.61	27.45 ± 1.15	196.43 ± 8.25	-	-	15.77 ± 0.66	12.99 ± 0.55
21.43	1-Isopropyl-4-Methylbicyclohexane-3-yl Acetate	1753/1750	*MS/RI*	3.31 ± 0.14	-	2.71 ± 0.11	-	-	-	-
Acids (2 kinds)										
8.966	Caproic Acid	1854/1843	*MS/RI*	0.57 ± 0.02	1.01 ± 0.04	0.85 ± 0.04	-	-	-	-
1.664	Acetic Acid	1490/1492	*MS/RI*	1.52 ± 0.06	1.69 ± 0.07	2.32 ± 0.1	3.18 ± 0.13	2.36 ± 0.1	-	-
Others (4 kinds)										
12.244	2-Ethyl-5-Methyl-Tetrahydro-Furan	1569/1568	*MS/RI*	0.28 ± 0.01	0.87 ± 0.04	-	-	-	-	-
28.026	Anethole	1812/1817	*MS/RI*	-	1.83 ± 0.08	1.24 ± 0.05	-	-	2.75 ± 0.12	-
20.74	Anthracene Violetin	1724/-	*MS*	0.61 ± 0.03	0.75 ± 0.03	-	-	-	-	-
51.463	Zanthoxylin	2676/-	*MS*	0.56 ± 0.02	0.79 ± 0.03	-	2.56 ± 0.08	-	2.11 ± 0.08	4.59 ± 0.19
24.115	Neryl Acetate	1728/1733	*MS/RI*	4.03 ± 0.17	2.35 ± 0.1	2.55 ± 0.11	3.96 ± 0.17	3.07 ± 0.13	-	-
23.69	Citronellyl Acetate	1657/1655	*MS/RI*	3.9 ± 0.16	2.75 ± 0.12	2.15 ± 0.09	3.43 ± 0.14	-	-	-
23.506	Terpinyl Acetate	1641/1652	*MS/RI*	-	-	-	28.57 ± 1.2	18.38 ± 0.77	-	-
sum of identified compounds (kinds)				54	53	47	49	37	19	16

Note: “-” means not identified; *MS* is qualitative mass spectrometry; *RI* is qualitative for the calculation of the retention index and is compared with the reference value in NIST general library. A, B, C, D, E, F, and G represent alcohols, aldehydes, ketones, hydrocarbons, esters, and acids, respectively.

## Data Availability

Data is contained within the article.

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
