# Peer review of "The Effects of Pepper (Zanthoxylum bungeanum) from Different Production Areas on the Volatile Flavor Compounds of Fried Pepper Oils Based on HS-SPME–GC–MS and Multivariate Statistical Method"

_molecules, 2022, doi:10.3390/molecules27227760_

Round 1
Reviewer 2 Report
Comments to the authors
General comments
The manuscript describes the strategy used in a research work aiming to determine a sort of fingerprint of different species of Zanthoxylum bungeanum from several production areas in China, in order to criate a flavor quality control technology. To do so, several compounds were selected, identified and data from them were collected. Chemometric tools were, then, applied to these data.
Quality control is always an important subject in every single field. This study, not only may help to create the mentioned data bank, but can also help people that work with “food bridiging” and “food pairing”, that have difficult to know which volatile compounds are available in a certain food, seasonings, aromatic herb etc.
Therefore, the information presented in the manuscript can be very helpful to improve food volatile flavor compounds data bank.
However, there is some improvement that could be made in order to help the reader to understand the authors reasoning and also to better extract information from the data.
After this improvement, in my opinion, the manuscript might be ready for publication.
Specific comments
It is important to say that the manuscript version I received had numbers of lines up to table 1. After this table these number disappeared from the manuscript, which made more difficult to situate the comment. I did my best to situate it and I hope I managed it.
Introduction
Lines 41-43: This sentence seems to be incomplete.
Figure 1: I may be mistaken but, according to Google maps, Shanxi Province is the province situated more to the northeast. The province displayed as Shanxi in this figure would really be Shaanxi Province. Could the authors, please, verify this information?
The authors selected Zanthoxylum bungeanum from 7 different regions in China. These regions were coded as S1 – S7. The correlation of theses 7 regions with this code S1 – S7 is made only in page 10 of the manuscript. However, since Table 1 (page 4) this correlation is important. Therefore, it can be very useful to have it stated in Figure 1 and in the text, as soon as these regions are presented.
Table 1: This is the main information source in the manuscript. Therefore, it has to exist and is very important. However, it is too large and, at least in my version, compound amount of some substances appears in more than 1 line. For example, for linalyl acetate and 1,5-dimethyl-1-vinyl-4-hexyl-2-aminobenzoate. The authors could either reduce slightly the font used, or reduce the number of decimals. I don´t think that it is necessary to express the amounts with so many digits. Standardize the number of the digits. There are different number of digits for the compound amounts.
Still in this table, the name of some compounds appear in three lines. Please, verify this situation and try to solve the problem.
Tables and figures have to be clear enough for the reader. Therefore, it is necessary to explain what was the uncertainty estimated for each compound amount. I guess the authors put the mean values and the standard deviation, but there are several forms to express uncertainty (confidence interval, one, two or three standard deviations, combined uncertainty (u(x)) or expanded uncertainty (U(x)). Please, let it clear not only in the table, but also in the text.
The second column in Table 1 represents a code for each compound. However, the authors did not use it in the manuscript. Every time it is necessary to mention the compounds, the authors write the whole name. Therefore, this column is useless. Either the authors remove it from the table to be able to expand the other columns and minimize the problems related here, or they use these codes along the manuscript.
The number for aldehydes that appear in brackets in Table 1 should be 10 and not 15.
The name of some compounds in Table 1 was written in uppercase, while others in lowercase. Please, standardize it.
Results and discussion
This part of the manuscript is very condensed. Discussion was poor and has to be improved. Most of the time the authors just describe what one can see from Table 1, but very few times try to interpret these results.
Lines 116-117 (I suppose): First paragraph after Table 1 – third sentence.
The authors say that the aromas detected from S1-S7 samples respectively were 52, 51, 45, 46, 35, 19 and 16. There is no such kind of numbers in Table 1. I tried to count them, but it did not match the text. Please, either use the code you put in the table or the name of the substances. This also makes very difficult to follow the fifth sentence of this paragraph.
Lines 117-119 (I suppose): First paragraph after Table 1 – fourth sentence.
In these sentences the authors mention several substances and say they are the common aroma components of Zanthoxylum bungeanum oils. And that´s the only information about theses components the authors put in the text. I think it could be more discussed. Theses amounts vary considerably among the species as it was put for components 16-52. I guess there could be a link between theses amounts with the characteristics of each substance etc, that was not even mentioned. These links will be corroborated, or not, by the chemometric tools used later in the manuscript. It is of no use to describe what one can see in a table if it cannot be discussed.
Lines 124-128 (I suppose): First sentence after Figure 2.
The authors mention 7 substances with relatively high content. However, trans-carvitol has a low content, if compared to the other 6. On the other hand, g-terpinol has much higher content and was not mentioned. I agree that this last one was detected only in oils S4 and S5, but it might be the reason for S4 and S5 oils show a different behavior from the others in HCA and PCA analysis, and, even if this is not the case, the authors could, at least, comment on it.
Lines 134-136 (I suppose): First sentence, second paragraph after Figure 2.
The authors say that 3 aldehydes have relatively high content (B1, B2 and B6). The same observation wasnmade for the alcohols (previous comment) can be made here. Two other aldehydes (B3 and B10) have approximately the same amount. These aldehydes were detected in S1, S2 and S3. Couldn´t it be possible that they contribute to the group observed in HCA ands PCA? It should be taken into account and later (after applying HCA and PCA). Explain whether they contribute or not and the reason for your hypothesis.
The same comments can be done for the other classes of substances lines 144-148 (hydrocarbons), lines 152-155 (esters), lines 159-162 (other substances). That is, compounds with more content are not mentioned or had their content discussed through their characteristics.
Sensory analysis
Lines 171-175: (I suppose) First paragraph after Figure 3 – first sentence.
S4 is put twice in this sentence. One time is enough.
Lines 174-177: (I suppose) First paragraph after Figure 3 – second sentence
It seems that there is something missing in the sentence. It can also be divided in two.
Still in this sentence, the authors say that S4 shows relatively strong trend all characteristics shown in Figure 3. These characteristics are correlated to limonene and linalool contents. However, S7 has the double amount of linalool content and does not show any strong trend related to any of the characteristics presented in the figure. Why is that?
Lines 177-184: (I suppose) First paragraph after Figure 3 – third sentence
In this sentence it appears the expression “differential differentiation”. It sounds odd. Could it be possible to change a bit this?
Still in this sentence the authors wrote that “it can be seen from Figure 1 that the relative content of hydrocarbon compounds in Sichuan Province red ... is the largest .... Zanthoxylum ... However, no amounts can be seen in Figure 1. Even if the figure number is uncorrect, for example if the authors change for Figure 2, still it will not be correct because in figure 2 the provinces are not displayed. Therefore, either the authors correct the sentence eliminating the mention of Figure 1 or the put the province code in the second figure and change the figure number in this sentence.
Lines 192-193 (I suppose)
The sentence in the manuscript says “this is consisted with the conclusions in Table 1”. Table 1 has no conclusions at all. There are numbers/data that have to be interpreted to lead to conclusions. Please, rewrite the sentence drawing these conclusions. It will certainly help the reader to follow your reasoning.
Cluster analysis
Line 207 (I suppose): Legend of figure 4
Why are practically all the words in upper case?
Lines 220-224 (I suppose):
Once more the authors mention that the difference among the oils can be explained by environmental conditions. However, they do did not tray to make any correlation of theses environmental conditions or the groups formed in Figure 4, for example. Is it possible to try to make any hypothesis about it?
Lines 225-235 (I suppose): First sentence of the second paragraph in page 11.
In this sentence, limone amount is said to be equal to zero in S3 oil. There is no analytical technique able to measure zero amount of any substance. What should have been written is “not detected”, as explained by the legend of Table 1. Please, change it in the manuscript.
PCA analysis
Figure 5
A PCA plot of scores and loadings could be more useful because could corroborate (or not) what the authors analysed till now from figures 3 and 4. That is, it could be possible to correlate province and flavor compounds. I suggest the authors to include in the text both graphics.
In general Figures 3, 4 and 5 were discussed poorly. Please, improve the interpretation of theses figures trying to make a link with flavor versus location.
Materials and methods
Lines 327-328 (I suppose): First sentence after formula 1.
Please, explain what do you mean by “quality” of an internal standard and “quality” of a sample.
Data analysis
Lines 370-372 (I suppose): The authors mention ANOVA analysis but did not show the results or explain the reason for the ANOVA application. Please, include this explanation in the text.
Conclusions
Most of what is written in the conclusion is not really a conclusion, but a repetition of the strategy applied in the study and the results obtained. What is left as a conclusion is very poor. Therefore is has to be rewritten and really conclusions should be put in this section of the manuscript.
Author Response
Dear expert: Please check the attachment

Reviewer 3 Report
The manuscript "Effects of Zanthoxylum bungeanum from different producing areas on volatile flavor compounds of fried Zanthoxylum bungeanum oils based on HS-SPME-GC-MS and Multivariate Statistical Method" describes the HS-SPME-CG-MS evaluation of the products from seven different areas. I have some concerns about the manuscript, as follows:
a) The manuscript must be totally revised. It is not an easy reading. There are excessive repetitions of the word "Zanthoxylum bungeanum" in all sections. The manuscript must not be accepted in the current form.
b) after my investigations into the field, I do not see novelty enough to justify the publication of this study. The authors should explain the main novelty of the current study when compared to other.
c) The number of samples is too low to support the conclusions.
d) What was the preprocessing method employed? The data treatment must be further detailed.
Author Response
Reviewer: 3
Comments to the Author
The manuscript must be totally revised. It is not an easy reading. There are excessive repetitions of the word "Zanthoxylum bungeanum" in all sections. The manuscript must not be accepted in the current form.
Response:We are truly grateful to you for reviewing our manuscript and giving us precious suggestions. Based on the reviewer’s comments and suggestions, we have revised the manuscript carefully (as can be seen in manuscript highlighted by red color). All the words "Zanthoxylum bungeanum" in the manuscript are changed to "pepper". In this paper, Dahongpao Zanthoxylum bungeanum in Hancheng County, Shaanxi Province, Dahongpao Zanthoxylum bungeanum in Yongshan County, Yunnan Province, Dahongpao Zanthoxylum bungeanum in Gangu County, Gansu Province, Hanyuan Red Zanthoxylum bungeanum oil in Sichuan Province, Dahongpao Zanthoxylum bungeanum in Mao County, Sichuan Province, Green Zanthoxylum bungeanum in Pingchang County, Sichuan Province, and Green Zanthoxylum bungeanum in Jiangjin District, Chongqing are modified to Hancheng pepper oil (HCPO), Yongshan pepper oil (YSPO), Gangu pepper oil (GGPO), Hanyuan pepper oil (HYPO), Maoxian pepper oil (MCPO), Pingchang pepper oil (PCPO), Jiangjin pepper oil (JJPO).
after my investigations into the field, I do not see novelty enough to justify the publication of this study. The authors should explain the main novelty of the current study when compared to other.
Response:We appreciate it very much for this good suggestion. The ecological environment has a significant impact on the aroma quality of pepper, and the volatile flavor components of pepper from different origins vary greatly. The food flavoring effect of pepper oil made from pepper of different origins varies significantly, however, there are few reports on the aroma research of pepper oil made from pepper of different producing areas, so it is necessary to study the volatile flavor components and key aroma substances of pepper oil of different producing areas. The manuscript describes the strategy used in a research work aiming to determine a sort of fingerprint of different species of Zanthoxylum bungeanum from several production areas in China, in order to criate a flavor quality control technology. To do so, several compounds were selected, identified and data from them were collected. Chemometric tools were, then, applied to these data. Quality control is always an important subject in every single field. This study, not only may help to create the mentioned data bank, but can also help people that work with “food bridiging” and “food pairing”, that have difficult to know which volatile compounds are available in a certain food, seasonings, aromatic herb etc. In this study, we used headspace solid-phase microextraction (HS-SPME) coupled with gas chromatography-mass spectrometry (GC-MS) to analyze the differences in volatile flavor components of pepper oil from different origins under the same process, The correlation analysis model of volatile substances among different pepper oil samples was constructed by principal component analysis, and the main volatile substances causing the flavor differences of pepper oil were initially clarified by combining with cluster analysis, and to conclude by descriptive sensory analysis that HYPO showed a strong trend in spicy aroma, fresh herbal aroma, floral and fruity aroma, fresh sweet aroma and fatty aroma. The results of the study provide data support for the establishment of an aroma composition mapping database for pepper oil, and provide theoretical and data references for the origin tracing of commercially available pepper oil. Therefore, this study has certain research value and novelty.
The number of samples is too low to support the conclusions.
Response:Special thanks to you for your valuable comments. When doing the study, We also thought of collecting more samples, but considered that the selected raw materials should have local characteristics for the following reasons. The planting area of pepper is mainly concentrated in East Asia, and Sichuan Province, Yunnan Province, Chongqing, Shaanxi Province, Shanxi Province and other places are the main producing areas of pepper in China. The seven kinds of pepper selected in this study are all local specialties and Chinese national geographical indication products, which are representative, with strong aroma, tasty and good quality, enjoying high reputation in the market, selling well both inside and outside the province, and loved by consumers.
Among them, Hanyuan pepper was listed as tribute in the Tang Dynasty, so they are called "tribute pepper". The color is red, the grains are large and oily, the aroma is rich, mellow and refreshing. It was awarded the title of provincial high quality product in 1984, and on February 4, 2005, the State Administration of Quality Supervision, Inspection and Quarantine approved the implementation of geographical indication product protection for "Hanyuan pepper"; Hancheng pepper has been cultivated for more than 600 years. On February 26, 2020, Hancheng pepper in Hancheng City, Shaanxi Province, was recognized as the third batch of Chinese characteristic agricultural products advantageous area; Yongshan pepper, raw green, hot red, pungent, numb, warm, fragrant, famous in the "Compendium of Materia Medica", has the effect of dilating blood vessels, strengthening the spleen and appetite, anti-cancer, etc. On August 2, 2006, the trademark was registered in Zhaotong Intellectual Property Office under the name of " Shante pepper king", its meaning is: special hemp, special oil, special fragrance; Gangu pepper, with a cultivation area of 200,000 mu, is the main source of income for local farmers, and in August 2022, the State Intellectual Property Office announced the list of national demonstration areas for the protection of geographical indication products in 2022. In August 2022, "Gangu pepper" geographical indications were successfully selected by the State Intellectual Property Office for the 2022 National Geographical Indication Product Protection Demonstration Area; Mao County pepper has a unique flavor of heavy oil, big grains, bright red color, strong fragrance, mellow hemp and delicious. Two "Mao County pepper Festivals" were held in Chengdu in 2008 and Dalian in 2009. "On December 31, 2010, the former State Administration of Quality Supervision, Inspection and Quarantine approved the implementation of geographical indication product protection for "Mao County pepper"; Pinchang pepper, with large and full grains, fresh color, rich fragrance and mellow taste, is not only a good seasoning product, its seeds can be squeezed for oil, and the peel, stalk, seeds and roots, stems and leaves can be used for medicine. On November 15, 2017, the former State Administration of Quality Supervision, Inspection and Quarantine approved the implementation of geographical indication product protection for "Pingchang pepper"; Jiangjin pepper is full of fruit, oily in color, fragrant and pure in taste. Eating it can increase appetite, which is the unique secret of Sichuan and Chongqing's spicy food. On October 17, 2005, the former State General Administration of Quality Supervision, Inspection and Quarantine approved the implementation of geographical indication product protection for "Jiangjin pepper". In November 2019, Jiangjin pepper was listed in the Chinese agricultural brand directory.
What was the preprocessing method employed? The data treatment must be further detailed.
Response:Thank you for your carefully reviewing. In our experiment, the quantification method is based on the internal standard method. 2-octanol is used as the internal standard compound, diluted to 4 μL/mL with anhydrous ethanol, added to the sample, and the concentration of each component is calculated based on the ratio of the peak area of the internal standard compound to the peak area of the component to be measured. Qualitative analysis: the isolated unknowns were matched with NIST / Wiley14 (only the identification results with a matching degree greater than 80% were selected); and the relative retention index (RI value) of the related compounds in the reference was compared and analyzed. Analysis of variance (ANOVA) was applied in the sensory evaluation of pepper oil. We applied the quantitative description method (QDA) to evaluate the sensory characteristics of seven pepper oils and established QDA plots for seven pepper oils. In order to analyze the difference of sensory characteristic intensity of pepper oil, SPSS 25.0 software was used to analyze the variance of the evaluated data.
Round 2
Reviewer 2 Report
Not all suggestions were implemented. However, the ones left out will not compromise the content.
Reviewer 3 Report
The manuscript was extensively revised and now can be accepted for publication.